# Thickness Measurements with EMAT Based on Fuzzy Logic

**DOI:** 10.3390/s24134066

**Published:** 2024-06-22

**Authors:** Yingjie Shi, Shihui Tian, Jiahong Jiang, Tairan Lei, Shun Wang, Xiaobo Lin, Ke Xu

**Affiliations:** 1Collaborative Innovation Center of Steel Technology, University of Science and Technology Beijing, Beijing 100083, China; b20200538@xs.ustb.edu.cn (Y.S.); b20200539@xs.ustb.edu.cn (S.T.); m202111260@xs.ustb.edu.cn (J.J.); m202221331@xs.ustb.edu.cn (T.L.); d202110589@xs.ustb.edu.cn (S.W.); 2Institute of Acoustics, Chinese Academy of Sciences, Beijing 100190, China; linxiaobo@mail.ioa.ac.cn

**Keywords:** electromagnetic acoustic transducer, thickness measurement, fuzzy logic, pulse compression

## Abstract

Metal thickness measurements are essential in various industrial applications, yet current non-contact ultrasonic methods face limitations in range and accuracy, hindering the widespread adoption of electromagnetic ultrasonics. This study introduces a novel combined thickness measurement method employing fuzzy logic, with the aim of broadening the applicational scope of the EMAT. Leveraging minimal hardware, this method utilizes the short pulse time-of-flight (TOF) technique for initial thickness estimation, followed by secondary measurements guided by fuzzy logic principles. The integration of measurements from the resonance, short pulse echo, and linear frequency modulation echo extends the measurement range while enhancing accuracy. Rigorous experimental validation validates the method’s effectiveness, demonstrating a measurement range of 0.3–1000.0 mm with a median error within ±0.5 mm. Outperforming traditional methods like short pulse echoes, this approach holds significant industrial potential.

## 1. Introduction

The Electromagnetic Acoustic Transducer (EMAT) is extensively employed in metal thickness measurements due to its non-contact operation, absence of couplant requirement, and high-temperature resistance [1]. However, the notably low conversion efficiency of the EMAT [2] necessitates high-performance excitation equipment and the amplification of weak signals. The low signal-to-noise ratio (SNR) primarily contributes to the complexity or even failure of signal processing. Despite these challenges, compared to piezoelectric ultrasonics, exploring the EMAT for thickness measurements remains a worthwhile research topic.

The accurate measurement of thickness presents substantial challenges when dealing with extremely thin or thick specimens, severe corrosion, high temperatures [3], or specimens with thick insulation layers [4]. As a result, considerable research has been undertaken to tackle these issues, with a focus on equipment [5], transducers [6], and methods for measuring thickness [7]. Companies like Ritec and Sonemat have notably contributed by creating linear power amplifiers, Class D power amplifiers, and related devices for the amplification of weak signals, offering invaluable assistance to research institutions. Some established commercial teams have developed narrowband handheld EMAT devices for thickness measurements, which exhibit stable performance, albeit within a conventional thickness range. Table 1 outlines the primary parameters of current EMAT thickness measurement equipment. In comparison, the fuzzy logic method introduced in this paper exhibits a broader measurement range and superior accuracy.

Regarding transducers [8], circular transducers are associated with spiral coils [9], while rectangular magnets correspond to racetrack or butterfly coils [10]. While these probes are indeed relatively advanced, significant potential remains for further exploration in the areas of transducer structure, coil layout, and magnetic field enhancement. For instance, Cai Z et al. [11] employed a longitudinal wave transducer and amplified the echo amplitude by optimizing parameters with the arrangement of permanent magnets in a Halbach array. Pei C et al. [12] introduced a modified meander-line-coil EMAT design with an innovative magnetic configuration aimed at increasing the peak flux density to enhance the generation and detection of surface waves. Kang L et al. [13] examined the effect of varying different EMAT parameters on surface waves using the orthogonal test method, resulting in a 25.2% increase in the signal amplitude for the optimized EMAT. Xu X et al. [14] assessed the impact of different coil shapes and excitation frequencies on the SNR of thickness measurement results and explored the operational range of the coil.

In terms of thickness measurement strategies, T. Takagi et al. [7] explored the impact of the Electromagnetic Acoustic Resonance (EMAR) method on thickness measurements in the context of steel corrosion thinning. S. Dixon et al. [15] utilized the conventional short pulse echo method to measure the thickness of low carbon steel and stainless steel under high-temperature conditions. Sheng-Bin Wang et al. [16] employed a pulse compression method to enhance the SNR. Zhao [17] used a laser source to generate shear waves and implemented electromagnetic ultrasonic receptions to measure the thickness of thin steel plates, approximately 0.4 mm thick, achieving impressive results. Qiu J et al. [18] managed to instantly excite a pulsed magnetic field by controlling the excitation current, which enabled the measurement of the thickness of 50 mm-thick steel plates with a small adsorption force.

EMAT equipment, transducers [19], and measurement methods have reached relative maturity. Current thickness measurement schemes mainly include the following: (1) building specific experimental systems for specimens’ thickness, and (2) using commercial thickness gauges. However, these methods require corresponding instruments and have their own limitations. Generally, the short pulse (Toneburst) echo method has a measurement range of 2–300 mm with moderate accuracy and is unable to measure thin or thick specimens beyond this range. Resonance methods can measure thin specimens with high precision but require large equipment and have high performance requirements, with few cost-effective commercial systems available. A linear frequency modulation echo is suitable for thick object measurement but requires wideband transmission and reception, with higher equipment demands and limited applicational scenarios. Hence, it is essential to explore a method to extend both the measurement range and accuracy under specific hardware conditions. Specifically, this study employs a strategy that utilizes the short pulse time-of-flight (TOF) method for initial thickness estimation, followed by secondary measurements guided by fuzzy logic principles. The integration of measurements from the resonance, short pulse echo, and linear frequency modulation echo extends the measurement range while enhancing accuracy. Therefore, the proposed method demonstrates potential for industrial applications.

This paper is structured as follows: Section 2 outlines various thickness measurement methods and analyzes limitations in constant hardware systems [20]. Based on the above analysis, Section 3 designs fuzzy logic and describes the thickness measurement process, which constitutes the main innovation of this study. Section 4 constructs an experimental system and validates it with 30 thickness samples, analyzing the measurement range and errors through fuzzy logic fusion, along with an analysis of measurement uncertainty. Section 5 concludes the paper, summarizing the findings and discussing potential avenues for future research.

## 2. Theory

### 2.1. Transduction Mechanism

The transduction mechanisms of the EMAT include the Lorentz force, magnetostrictive force, and magnetization force. Among them, the Lorentz force is primarily applicable to non-magnetic materials, while both the Lorentz force and magnetostrictive force are applicable to magnetic materials. A permanent magnet generates a static magnetic field $B_{s}$ in the direction perpendicular to the specimen, and an alternating current $J_{c}$ is passed through the coil, creating a time-varying magnetic field $B_{d}$ around the coil. This induces an oppositely directed eddy current $J_{e}$ in the sample. The dynamic magnetic field $B_{d}$ flows horizontally on the surface of the specimen in the eddy current region. As a result, the eddy currents $J_{e}$ produce a horizontal Lorentz force $F_{s}$ and a vertical Lorentz force $F_{d}$ under the influence of the static and dynamic magnetic fields, respectively. Additionally, in magnetic materials, the magnetostrictive forces $F_{M}$ are horizontal vibrations. Figure 1 shows a schematic diagram of these transduction mechanisms.

In this thickness measurement application, since the amplitude of the static magnetic field is much greater than that of the dynamic magnetic field, the designed EMAT primarily relies on the static Lorentz force. The specimen is subjected to a force in the horizontal direction within the range of skin depth, while the propagation direction of ultrasonic shear waves is vertical. The transduction processes described above can be represented by the following equations. The volume force of the Lorentz $F_{s}$ force arises according to Equation (1). Equation (2) describes the strain generated by the magnetostrictive effect, where *ε* represents the strain tensor, *σ* represents the stress tensor, and *H* represents the magnetic field intensity.
(1)$$F_{s}=J_{e}\times B_{s}$$
(2)$$\varepsilon =f\left(\sigma ,H\right)$$

Equation (3) illustrates that the reception of ultrasonic waves is the reverse process of the generation of static Lorentz forces. The particles cut the magnetic inductance lines, creating an induced electromotive force. For a continuous length circular coil, it can be expressed by the following equation, where *J*_r_ represents the total induced current received in the coil, *μ* represents the particle displacement, *η* represents the electromagnetic conversion efficiency, *σ* represents the electrical conductivity of stainless steel, and *l* represents the integral length of the coil. Equation (4) represents the inverse magnetostrictive effect, which is the process of stress generating magnetic induction, where *B* represents the magnetic flux density.
(3)$$Jr=\int_{0}^{L}\eta \sigma \left(\frac{𝜕\mu }{𝜕t}\right)\times B_{d}dl$$
(4)$$B=y\left(\sigma ,H\right)$$

### 2.2. TOF Measurement

When ultrasonic waves encounter an interface, they are reflected. Given the speed of sound, the distance that the ultrasonic waves travel within the specimen can be calculated by measuring the time from emission to reception. This process is known as time of flight (TOF). The specimen thickness can be calculated using the following formula, where $c_{s}$ represents the shear wave velocity, d represents the thickness, $\Delta t_{i}$ represents the time difference between the *i*-th and (*i*-1)-th echoes, and N represents the average number of valid echoes:(5)$$d=\frac{\sum c_{s}\Delta t_{i}}{2N}\;\;\;\;\;\;\;(i=1,2,\ldots N)$$

Continuous wave (CW) and linear frequency modulation (LFM) are commonly used as excitation waveforms, as depicted in Equations (6) and (7), respectively. The frequency of the CW signal is $f_{c}$, while A, $f_{0}$, B, and τ represent the amplitude, starting frequency, bandwidth, and pulse width of the LFM signal, respectively. Depending on the number of waves (M) in the pulse train, CW signals can be divided into short pulse and long pulse, which are used for thickness measurements by the TOF and resonance method, respectively. Typically, Toneburst signals consist of fewer than 5 waveform cycles, and windowing is applied. In contrast, resonant sweep signals contain a large number of cycles.
(6)$$u_{\mathrm{cw}}\left(t\right)=A*\sin\left(2\pi f_{c}t\right)\;0\leq t\leq M/f_{c}$$
(7)$$u_{\mathrm{in}}\left(t\right)=A*\sin\left(2\pi \left(f_{0}t+\frac{Bt^{2}}{2\tau }\right)\right)\;0\leq t\leq \tau$$

The basic processing flow of the echo method algorithm includes the following: noise reduction [21], pulse compression (which is a specific method for LFM) [22], envelope extraction, and the peak detection of echoes [3], as well as the calculation of average time intervals between multiple peaks.

### 2.3. Resonance Measurement

The EMAR [23] utilizes the principle of echo superposition and employs a sweep frequency method for excitation. When the resonance enhancement frequency corresponds to the thickness, the spectral response sharply protrudes. The thickness of the specimen can be derived by calculating the frequency difference between adjacent spectral peaks, where $f_{N}$ represents the frequency of the *N*-th harmonic peak.
(8)$$d=\frac{\sum \frac{c_{s}}{2\left(f_{N+1}-f_{N}\right)}}{N-1}\;\;\;\;\;\;\;\;\;\;\;\;(i=1,2,\ldots (N-1))$$

Equation (8) is employed to determine the thickness of the specimen. The thickness is computed by measuring the harmonic frequencies of various orders, obtaining frequency differences, and calculating the thickness from these values. Contrary to the analog heterodyne spectrum scanning method used in RITEC-5000 equipment, this study employs ADC to directly acquire the original signal. Through a frequency domain analysis, the amplitude of the corresponding frequency at each sweep point is obtained. Full digital processing is also a part of this work.

It should be noted that the excitation length is generally the same, and the signal is subjected to an FFT analysis for an equal duration after the blind zone. It shows resonance achieved at 3.36 MHz with a 30 μs excitation for a 1.4 mm-thick aluminum plate. Figure 2a,b depict the time–domain and frequency–domain results, respectively. Since the maximum blind zone in the measured sweep occurs around 30 μs after the signal ends, this study uses 163.84 μs (8192 points) of data starting from 70 μs for the FFT analysis. The amplitude of the energy at the frequency axis of 3.36 MHz is 1.56, which represents a single resonance amplitude value. By performing a sweep from 0.5 MHz to 5 MHz with a 39.6 kHz interval, the sweep results shown in Figure 2c can be obtained.

### 2.4. Limitations and Solutions

The main limitation of current Toneburst-based EMAT thickness measurement systems is their limited range, leading to reduced accuracy and stability beyond this scope. As the thickness of the specimen decreases, it becomes difficult to accurately distinguish the time intervals between echoes in the time domain using TOF [24], which is mainly due to the following two aspects:

(1) The electronic system: The duplex circuit between the EMAT high-voltage pulse transmission and weak signal amplification results in a system blind spot. When the time interval between echoes is short, the echo with a higher signal intensity mainly exists in this time period, which decreases the SNR. Due to the existence of the system blind spot, it may not be possible to accurately detect the echo signal.

(2) The ultrasonic propagation distance and wavelength: As the thickness of the specimen decreases, the ultrasonic propagation distance approaches or becomes smaller than the transmitted wavelength, resulting in the superposition of multiple echoes in the time domain. When multiple echoes overlap, the temporal characteristics of the echo signal change, making it difficult to resolve and measure the echoes.

In Figure 3a, the blue dashed box indicates the saturation state. The filter and oscillation attenuation circuit mitigate the superposition of oscillations at specific frequencies, thereby reducing the blind zone. In Figure 3b, three echoes are present. The short intervals between echoes lead to an overlap within the blue box. The uncertain superposition effect, due to the correlation between the echo time and wavelength, hinders the differentiation of echoes through signal processing.

In any EMAT system, these issues are unavoidable, and special attention needs to be paid to these limitations in the case of thin thickness measurements. To overcome these issues, other methods or techniques may need to be adopted, such as using higher frequencies, improving circuits, or combining other measurement methods.

In this study, the authors propose the use of the resonant method [25] on existing hardware to solve the problem of thin specimen thickness measurements and reduce the blind zone. The resonant method requires at least one spectral peak to fall within a flat transmission–reception frequency range to ensure valid thickness measurement results. The ZYNQ board is used for DAC waveform generation and ADC sampling. The power amplifier and weak signal amplifier are limited in frequency bands due to electronic equipment, with the main frequency range shown in Table 2. The flat frequency range is designed to be 0.5 MHz to 5 MHz due to the influence of parameters of the electronic system and coil. By employing Equation (8), the theoretical minimum thickness is ascertained to be 0.323 mm. It should be noted that, under identical conditions of the frequency scanning range and resolution, an increase in thickness results in a decline in measurement precision; thus, the resonance method is ideally suited for gauging the thickness of thin specimens.

When the specimen thickness is large, the increased propagation distance of ultrasonic waves results in significant signal attenuation, which poses a challenge, especially for EMAT systems with a low SNR. Traditional methods, such as increasing the number of CW signals, cannot simultaneously enhance the signal intensity and distance resolution capability. In this case, using LFM signals is a better choice. By selecting an appropriate bandwidth and pulse width for linear frequency modulation signals, the signal intensity and distance resolution capability can be increased. The broadband nature of LFM allows them to propagate over a larger thickness range and provides better signal intensity. Additionally, pulse compression processing can further improve the temporal resolution and peak amplitude of the signals. Therefore, using LFM signals can achieve better performance in measuring the thickness of thick specimens, enhancing the signal intensity and distance resolution and achieving accurate thickness measurements.

Based on the above analysis, the three thickness measurement methods differ in range: the EMAR method is better suited for thin pieces [26], Toneburst for medium thicknesses, and LFM for thick measurements. Under limited hardware conditions, this paper employs fuzzy logic to integrate these three methods, autonomously selecting the appropriate measurement method, thereby broadening the range.

## 3. Fuzzy Logic Measurement

### 3.1. Fuzzy Logic Design

Fuzzy logic [27], as a mathematical tool, is used to address the uncertainty and ambiguity present in problems. Data fusion, based on fuzzy logic, can resolve differences in attributes from various data sources [28]. Given that different sensors or measurement methods have varying resolutions, ranges, and accuracies, a more comprehensive and accurate dataset can be obtained through fuzzy logic fusion [29]. Fuzzy logic is applicable to the data fusion of various EMAT thickness measurement methods; hence, it is introduced in this paper.

Based on the theoretical analysis in Section 2, the extensive experimental data in Section 4, previously published papers, and the range intervals provided by commercial thickness measurement devices, this paper concludes that the resonance method has an accuracy range of 0.3–4.0 mm, the Toneburst method ranges from 2.0 to 150.0 mm, and LFM exceeds 20.0 mm. These estimates are based on experience. Considering the uneven distribution of the measurement range and the requirement for values to start from 0, the normalized value $D_{i}$ is obtained as shown in the following formula.
(9)$$D_{i}=\log2\left(d_{i}+1\right)$$

In the above formula, $d_{i}$ represents the thickness (in mm) measured by different methods. The relationship between $D_{i}$ and ${(d}_{i}+1)$ is logarithms, ensuring a uniform distribution of each thickness measurement method along the X-axis. This design rationale is demonstrated in Figure 4, Figures 10 and 11. Introduce the average value D of the effective normalization results of three methods as per the following formula.
(10)$$D=\frac{\sum_{i=1}^{N}D_{i}}{N},\;N=1,2,3.$$

In the above formula, N represents the total number of effective measurement methods at this thickness. For example, both the EMAR and Toneburst method can effectively measure an aluminum plate of 1.201 mm in thickness. In this scenario, $d_{1}=1.216$, $d_{2}=1.175$, N=2, $D_{1}=1.148$, $D_{2}=1.121$, and D=1.1345.

The membership function used in the fuzzy rules is a Gaussian function, and the equation is as follows.
(11)$$k_{i}=e^{-\alpha_{i}{\left|D_{i}-X_{i}\right|}^{2}},i=1,2,3.$$
where $\alpha =4/\gamma^{2}$, γ is a constant determining the distribution of the rule. The γ values selection principle ensures good superposition between different logical determinations and is adjusted based on the thickness measurement ranges of different methods. In this section, the α values used are all 1.2. $X_{i}$ represents the median value in the design, which is, respectively, taken as 0, 4, and 6 in the three methods. The value of $X_{i}$ is determined by the measurement range of different methods, which is typically chosen at the center of a well-defined range. Based on experimental data from existing test blocks, this paper selects the thickness center as the value of $X_{i}$, ensuring maximum weighting at the center. Specifically, for the EMAR method, the center is set at 0 mm, so $X_{1}=D_{1}\left(0+1\right)=0$. For the Toneburst method, $d_{2}$ is set to 15, corresponding to the center of effective specimens ranging from 2 to 150 mm, and $X_{2}=D_{2}\left(15+1\right)=4$. Take $X_{3}=6$ as an integer. The choice of $X_{i}$ can be based on either theoretical or experimental measurement ranges. This paper determines the fuzzy logic parameters through limited experimental data to achieve better results.

The Gaussian function is normalized to obtain the normalized weight values, and this is to ensure that the result is true.
(12)$$c_{i}=\frac{k_{i}}{\sum_{1}^{3}k_{i}},i=1,2,3.$$

Figure 4 displays the membership function, with the horizontal axis representing D on a logarithmic scale. The vertical axis represents the amplitude of the normalized weight $c_{i}$. The red, blue, and green curves in this figure represent the measurement result weights of the resonance method, the Toneburst method, and the LFM method, respectively.

The fuzzy logic [30] rules are presented in Table 3, which is categorized into small (S), small medium (SM), medium (M), medium large (ML), and large (L).

The calculation method for the thickness d varies when the normalized thickness D belongs to different rules, where $R_{1}$ represents the first rule, and so on. For instance, when D=1.1345, the fuzzy logic complies with the first rule and is determined as S. In this case, $d=d_{1}=1.216$ mm. When the nominal thickness value is 6.0 mm, D=2.6411, and it is determined as SM. At this time, according to Formulas (9)–(12), $c_{1}=0.0021$, $c_{2}=0.9979$, and d=0.0021∗5.437+0.9979∗6.033=6.032 mm.
(13)$$\left\{\begin{array}{c} R_{1}:IF\left(D\in S\right)\;\;\;\;\;\;\;\;\;\;\;\;\;\;\;\;\;\;\;\;\;\;\;\;\;\;\;THEN(d=d_{1})\\ R_{2}:IF\left(D\in SM\right)\;\;\;\;\;\;THEN(d=c_{1}d_{1}+c_{2}d_{2})\\ R_{3}:IF\left(D\in M\right)\;\;\;\;\;\;\;\;\;\;\;\;\;\;\;\;\;\;\;\;\;\;\;\;\;\;THEN(d=d_{2})\\ R_{4}:IF\left(D\in ML\right)\;\;\;\;\;\;\;\;\;\;\;\;\;\;THEN(c_{2}d_{2}+c_{3}d_{3})\\ R_{5}:IF\left(D\in L\right)\;\;\;\;\;\;\;\;\;\;\;\;\;\;\;\;\;\;\;\;\;\;\;\;\;\;\;THEN(d=d_{3}) \end{array}\right.$$

The above design of fuzzy logic rules is abstract. To help readers grasp this concept more vividly, this section provides illustrative examples by substituting partial data from Table 4 in Section 4 into selected formulas.

### 3.2. Principles of Fuzzy Logic and Thickness Measurement Process

The thickness measurement procedure, as illustrated in Figure 5, is described as follows:

(1) The measurement process begins with the excitation of a 3-cycle 4 MHz pulse. The echo signal processing algorithm is detailed in Section 2.

(2) Based on the initial thickness measurement results and fuzzy logic, a secondary measurement method is selected: (a) For thicknesses less than or equal to 10 mm, or those that are undetectable, the resonant method is applied. A frequency sweep is conducted at 39.6 kHz intervals within the 0.5 MHz to 5 MHz range. The frequency difference is calculated from the envelope of the spectral response curve to determine the measured thickness. (b) For initial thickness measurements greater than or equal to 20 mm, or those that are undetectable, the LFM method is used.

(3) Fuzzy logic is applied to merge the secondary measurement results, yielding the final thickness measurement.

## 4. Experiments

### 4.1. Experimental Setup

The experimental setup, as depicted in Figure 6, encompasses a variety of components including a ZYNQ board, a power amplifier, a weak signal amplifier, a duplexer, and an EMAT. At the core of the experimental setup is the ZYNQ7020 (Xilinx, San Jose, CA, USA) board, a specialized circuit designed around a heterogeneous FPGA. This board generates TTL Gate and Sin signals through DAC902 (TI, Dallas, TX, USA) and utilizes AD9226 (ADI, Wilmington, MA, USA) for sampling, with a refresh rate of 50 MHz. The entire system operates on a Linux platform running Jupyter and employs Python for programming. It controls the transmission of signals to the transducer through a power amplifier while concurrently capturing the incoming analog echo signals. The Ga2500 (RITEC, Warwick, RI, USA) power amplifier can deliver up to 5 KW of power output. Additionally, the signal amplifier, specifically RITEC’s BR640A (RITEC, Warwick, RI, USA), comes equipped with a hardware filter ranging from 100 kHz to 5 MHz and offers a gain of 68 dB. The duplexer was meticulously designed to achieve broadband impedance matching with the probe.

The specimens for the experiment included a variety of samples such as thin aluminum plates, carbon steel step blocks, long carbon steel bars, and other types. These samples, with thicknesses ranging from 0.2 to 1000.0 mm, were measured in accordance with the study’s requirements. This diverse selection of specimens allowed for a comprehensive and robust evaluation of the system’s performance.

### 4.2. Experimental Verification

Drawing upon the theoretical analysis in Section 2.4, this paper conducted thickness measurements on thin aluminum plates ranging from 0.2 mm to 10 mm using the resonance method. The echo waveform is shown in Figure 5. The thickness values were calculated according to the method outlined in Section 2.3 and recorded in Table 4.

The partial results of the resonance scan are illustrated in Figure 7, where 0.3, 0.5, 0.8, 6.0, 8.0, and 10.0 mm are plotted, representing the minimum and maximum cases, respectively. At the nominal value of 0.3 mm (at the system’s measurable bandwidth boundary), the resonance fundamental frequency is 5 MHz, resulting in a calculated thickness of 0.323 mm. It can be observed that the thickness measurement accuracy is poor at this limit. At the nominal value of 0.5 mm, the resonance fundamental frequency is 3.210 MHz, yielding a calculated thickness of 0.503 mm, where the error is small. For a large thickness value of 6 mm, the average frequency difference of adjacent resonance harmonics is (2.140 − 1.546)/2 = 0.297 MHz, giving a thickness value of 5.437 mm. With a frequency difference of 0.278 MHz (obtained from 1.824 MHz minus 1.546 MHz), the corresponding thickness is calculated to be 5.809 mm. It can be seen that the thickness measurement error is larger when the thickness is larger, which aligns with the theoretical analysis. This is primarily due to the increase in the ratio of frequency resolution to frequency difference. If the resonance method is still used for thickness measurements at this time, the accuracy is significantly compromised.

Measurements of varying thicknesses (12.0, 24.0, 36.0, …, 155.0, and 200.0 mm) were conducted using the Toneburst method, as depicted in Figure 8. As the propagation distance increased, the amplitude of the echo signal diminished. For example, at 155 mm, it became challenging to discern the secondary echo. In the case of larger thicknesses, the short pulse echo method’s limitation becomes apparent, as errors cannot be mitigated by averaging the intervals between multiple echoes.

Taking a 100.0 mm test block as an example, thickness is measured using the LFM method. Unlike the Toneburst method, which involves multiple acquisitions and time–domain averaging, the pulse compression of a single acquisition datum from Figure 9a yields Figure 9b. The thickness value of 100.13 mm is calculated based on the time difference of 62.0 µs between two adjacent reflected pulse peaks (186.0 − 124.0 = 62.0 µs). Compared to the single acquisition in the time domain of Figure 8, the SNR is significantly improved. To observe effective signals, the time axis is set to 100–250 µs.

### 4.3. Fuzzy Logic Fusion

The thickness of 30 test samples was individually measured. For instance, if the EMAR and Toneburst method are applicable to the 2.0 mm test sample, these methods are used to measure the effective thickness value ten times, respectively. The averages of these measurements, 2.038 mm and 2.024 mm, respectively, are recorded in Table 4.

The second column of the table displays the thickness indicated at the time of purchase for each specimen. However, some blocks are not standard. To validate the “actual thickness” against the EMAT measurement results, a high-precision micrometer is employed for calibrating test blocks of 10 mm and below, while vernier calipers are used for blocks larger than 10.0 mm. The thickness tolerance of the workpiece is minimized by averaging five measurements, as recorded in column 3. Columns 4, 5, and 6 present thickness measurements obtained from three distinct methods. A dash “-” signifies that the method was not utilized. The seventh column represents the fuzzy logic determination based on the principles outlined in Table 3, while the eighth column records the final thickness measurement result, derived from the fusion of different methods via fuzzy logic. The ninth column represents the error between the thickness measurements obtained using fuzzy logic and those obtained with a micrometer.

The experimental results show that the resonance method is only effective for measuring relatively small thicknesses. The traditional echo method has a blind spot for small thicknesses and cannot measure large thicknesses. The LFM method can measure thick samples and demonstrates better accuracy and range.
sensors-24-04066-t004_Table 4Table 4Thickness measurement results of three methods and fuzzy logic fusion.MaterialThickness (mm)Micrometer*d* (mm)EMAR *d_1_* (mm)Toneburst *d_2_* (mm)LFM *d_3_* (mm)Fuzzy JudgmentFuzzy Thickness Result (mm)Fuzzy Error (mm)Aluminum0.20.198----0.000-0.30.3120.323--S0.3230.0110.50.5040.503--S0.503−0.0010.80.8030.7990.632-S0.799−0.0041.00.9960.9970.984-S0.9970.0011.21.2011.2161.175-S1.2160.0151.41.3941.4021.416-S1.4020.0081.51.4831.4781.365-S1.478−0.0052.02.0352.0382.024-SM2.0380.0033.03.0033.0072.985-SM2.996−0.0074.04.0294.0494.001-SM4.003−0.0265.05.0304.9245.048-M5.0480.0186.06.0455.4376.033-M6.033−0.0128.08.0047.9668.001-M8.001−0.00310.010.03710.18810.063-M10.0630.026Carbon steel12.012.00-11.92-M11.920−0.0815.014.98-14.92-M14.920−0.0619.019.04-19.12-M19.1200.0824.023.98-23.9523.97M23.950−0.0330.030.02-29.8929.96ML29.921−0.09936.036.00-35.9636.00ML35.989−0.01142.042.00-42.0242.01ML42.0110.01148.048.00-48.0348.02ML48.0200.0280.080.04-80.0980.03L80.030−0.01100.0100.06-100.04100.08L100.0800.02155.0155.69-156.30155.99L155.9900.3200.0200.32-201.40200.21L200.210−0.11400.0401.90--401.56L401.560−0.34600.0598.30--599.04L599.0400.741000.01000.40--1000.17L1000.170−0.23


In the above table, 0.2 mm falls outside the measurement range, rendering the actual result incorrect. The measurement range of different methods is depicted in Figure 10, with the X-axis representing the logarithm of the actual thickness. The Y-axis heights represent the four distinct methods, with each method corresponding to a specific height. The presence of marks signifies the effective measurement of thickness using these methods. The figure illustrates that the three measurement methods have overlapping and complementary ranges. The fuzzy logic thickness measurement results encompass the entire measurement area, thereby enhancing the thickness measurement range.

The comparison of various thickness measurement methods is presented in Table 4, with corresponding errors illustrated in Figure 11. This figure displays the errors of the EMAR, Toneburst, LFM, and fuzzy methods, represented by red, blue, green, and magenta lines, respectively. Generally, single-method measurements exhibit substantial errors at the range extremes. Specifically, (1) the Toneburst method demonstrates poor accuracy and stability for measurements below 2.0 mm, with errors reaching 21.3% at 0.8 mm. Conversely, the EMAR method performs well within this range, and fuzzy logic accurately categorizes it as “small”. (2) The EMAR method exhibits significant errors beyond 5.0 mm, attributable to a single sweep interval. However, in the overlapping region with Toneburst, fuzzy logic designates it as “M”, yielding accurate measurements. (3) The Toneburst method incurs substantial errors at or beyond 155 mm and fails to measure thicknesses at 400.0, 600.0, and 1000.0 mm. To extend the measurement range for thicker samples, the LFM method is introduced. Fuzzy logic categorizes this range as “ML” or “L”.

In summary, the integration of different methods’ measurements through fuzzy logic reduces errors, expands the measurement range, and enhances robustness.

### 4.4. Measurement Uncertainty

The integration of thickness results via fuzzy logic extends the measurement range. This approach compensates for errors inherent in a single method, thereby reducing measurement inaccuracies. Uncertainties in thickness measurements arise from various factors, including electronic systems and temperature fluctuations. Errors due to the electronic system are quantified and offset by a calibration block, the details of which are not discussed here. To maintain temperature stability, the test blocks were conditioned at room temperature for 24 h prior to experimentation. Each of the 30 types of specimens underwent 10 fuzzy logic thickness measurements, followed by an analysis of measurement uncertainty. Figure 12 illustrates the thickness measurement error, where the X-axis represents the sequence number of different thicknesses (i.e., 1, 2, 3…, 30), and the Y-axis denotes the error (in mm) between the fuzzy logic measurement result and the actual thickness (*d*).

The figure above demonstrates that the median error for varying thicknesses remains within ±0.5 mm, with all errors falling within ±1.2 mm. This error appears to increase proportionally with thickness, a trend primarily attributed to the differences in thickness measurement methods and the thickness of the specimen itself. For smaller thicknesses, the errors are not only minimal but also tightly clustered, a trend that underscores the stability and high accuracy of the EMAR method. Conversely, larger thicknesses are associated with larger errors and a greater number of outliers, some of which exceed 1 mm. Despite the magnitude of the error increasing with thickness, the error percentage remains relatively small. This discrepancy could potentially be explained by two main factors: (1) the accuracy and range limitations of the vernier caliper and (2) the reduction in the number of echoes within the same sampling time, which weakens the average effect of multiple echo intervals.

## 5. Conclusions and Future Work

For hardware with a limited frequency bandwidth, different methods exhibit varying thickness measurement ranges and accuracies. By integrating fuzzy logic with the resonance, short pulse echo, and LFM methods, this paper has designed a thickness measurement approach. This approach effectively extends the measurement range, enhances accuracy, and strengthens the robustness of measurements. The proposed approach of a primary conventional evaluation and a secondary fuzzy logic thickness measurement scheme can effectively improve the measurement efficiency. The experimental results demonstrate that by using fuzzy logic, this study achieves a large measurement range of 0.32–1000 mm with an accuracy of 1%, providing a direction for enhancing the performance of existing thickness measurement devices.

Future work will focus on four key areas. (1) Frequency selection: different thickness measurement methods correspond to different receiver frequency bands. By incorporating a switchable frequency selection circuit, the signal-to-noise ratio of the echo signal can be enhanced when using a single method. (2) Data accumulation and fuzzy logic: as thickness measurement data accumulates, prior knowledge can be utilized to refine and enhance the fuzzy logic. (3) Equipment miniaturization: based on existing work, the power amplifier and preamplifier should be miniaturized to achieve overall customization of the equipment. (4) Utilizing accumulated thickness data, this research can explore the performance of lightweight AI algorithms and their potential deployment on small, embedded platforms in the future.

## Figures and Tables

**Figure 1 sensors-24-04066-f001:**
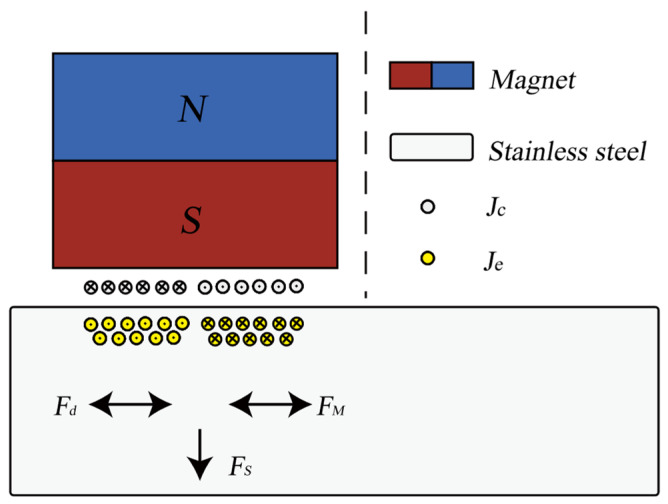
A schematic of the EMAT transduction mechanisms.

**Figure 2 sensors-24-04066-f002:**
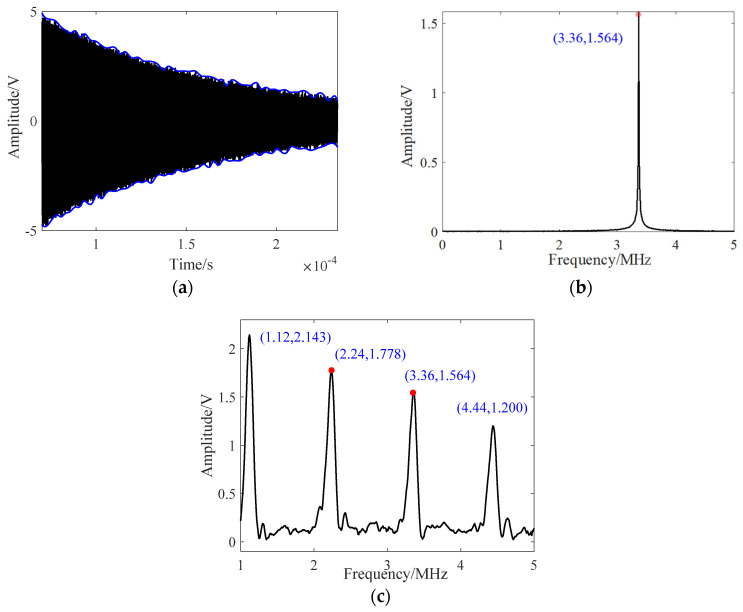
The digital resonance analysis: (**a**) time domain, (**b**) frequency domain, and (**c**) the sweep results.

**Figure 3 sensors-24-04066-f003:**
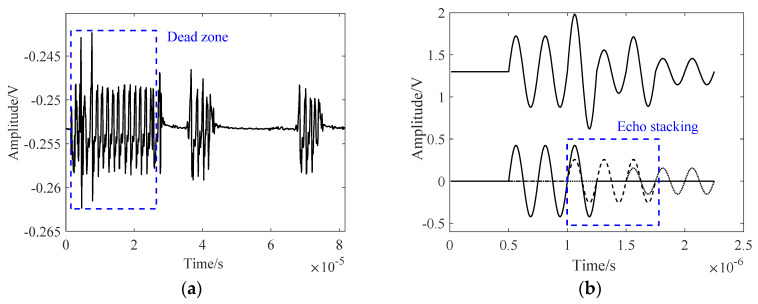
Reasons for failure of the TOF method: (**a**) blind zone and (**b**) echo overlap.

**Figure 4 sensors-24-04066-f004:**
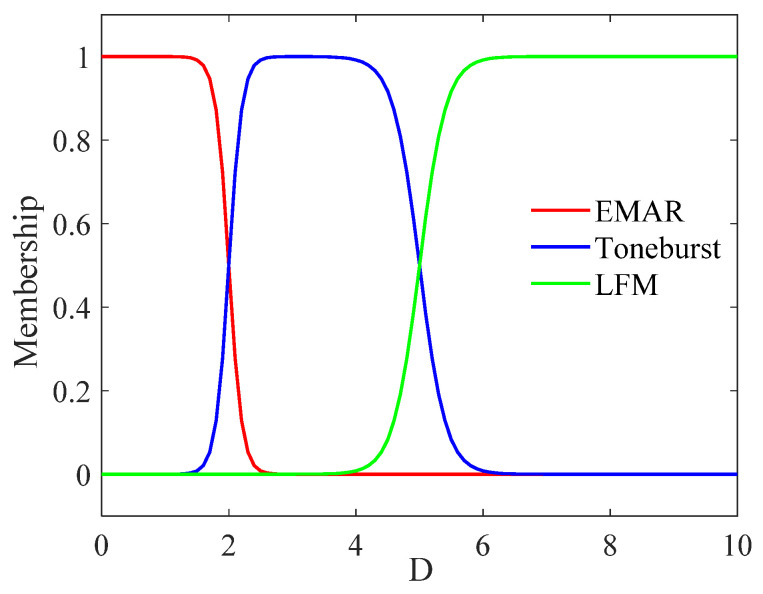
Membership function diagram.

**Figure 5 sensors-24-04066-f005:**
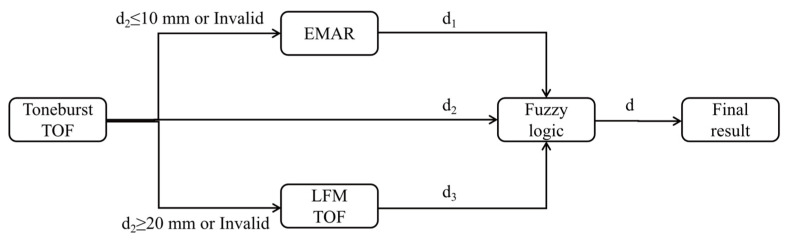
Flow chart of the thickness measurements.

**Figure 6 sensors-24-04066-f006:**
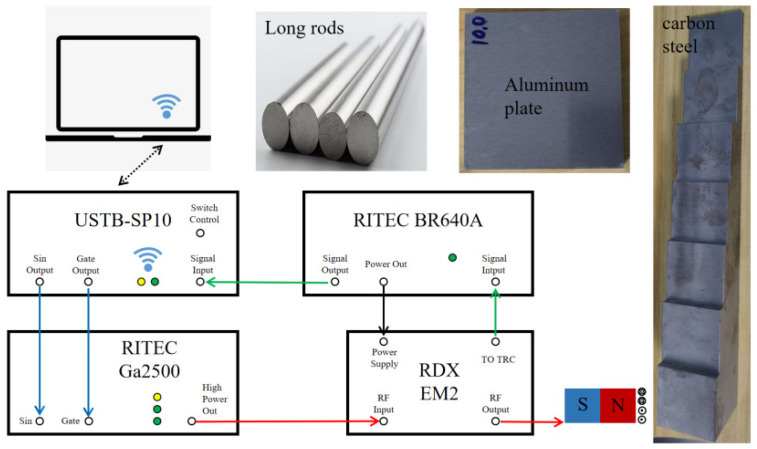
Diagram of experimental system.

**Figure 7 sensors-24-04066-f007:**
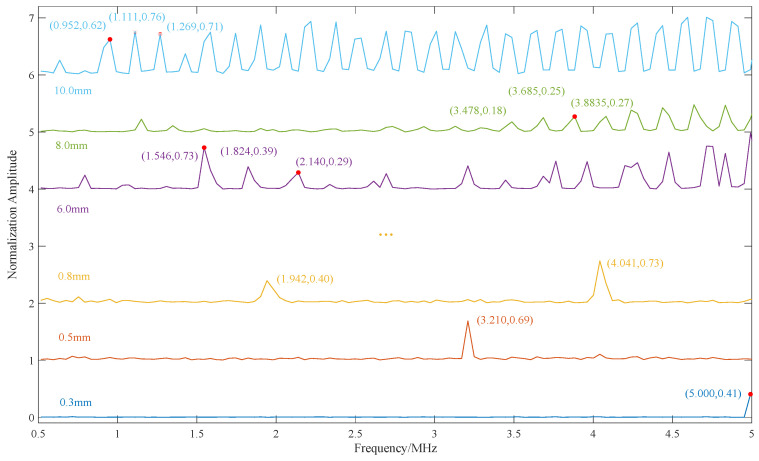
Resonance scanning.

**Figure 8 sensors-24-04066-f008:**
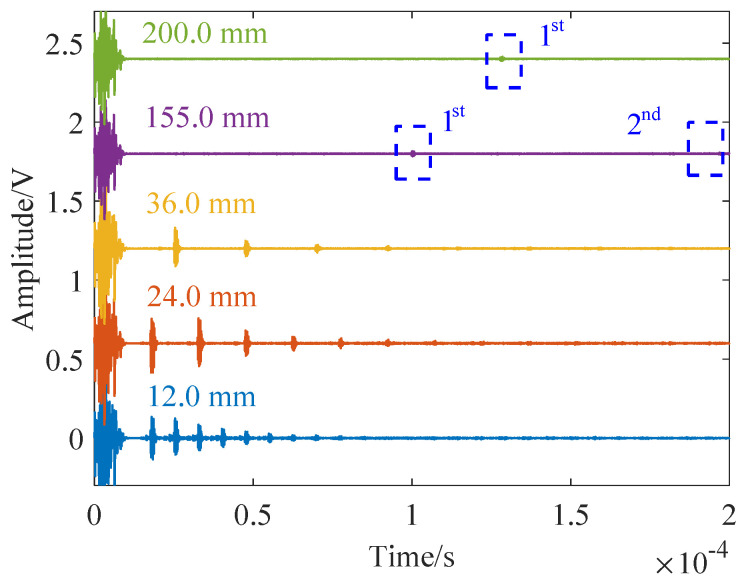
Thickness measurements using Toneburst pulses.

**Figure 9 sensors-24-04066-f009:**
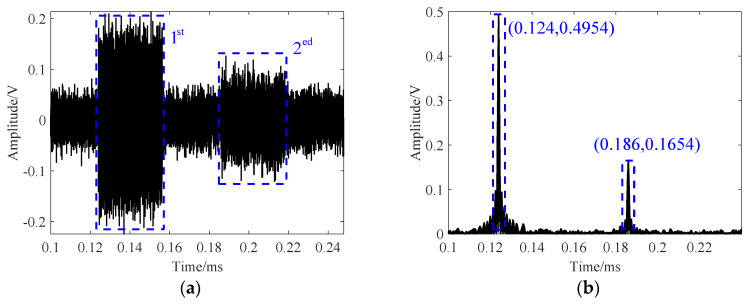
Thickness measurements using the LFM method. (**a**) Time–domain echo. (**b**) Results of pulse compression.

**Figure 10 sensors-24-04066-f010:**
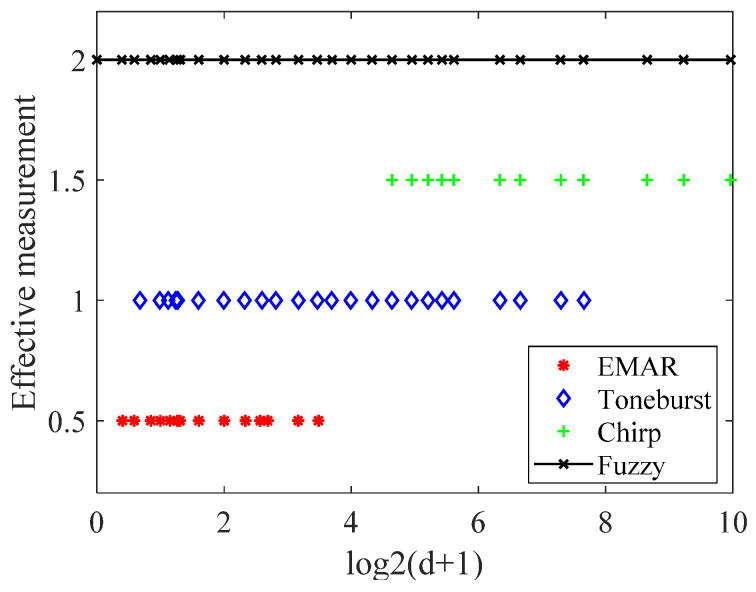
The range of different methods.

**Figure 11 sensors-24-04066-f011:**
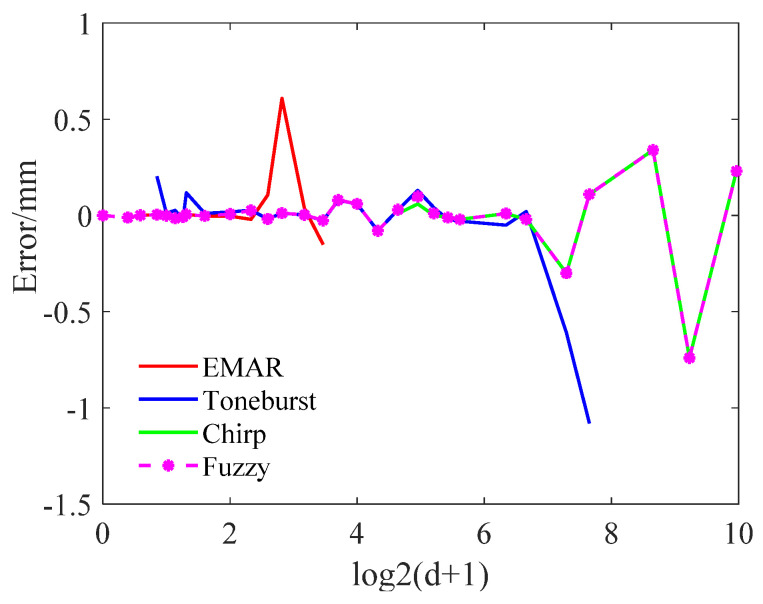
Error analysis of different methods.

**Figure 12 sensors-24-04066-f012:**
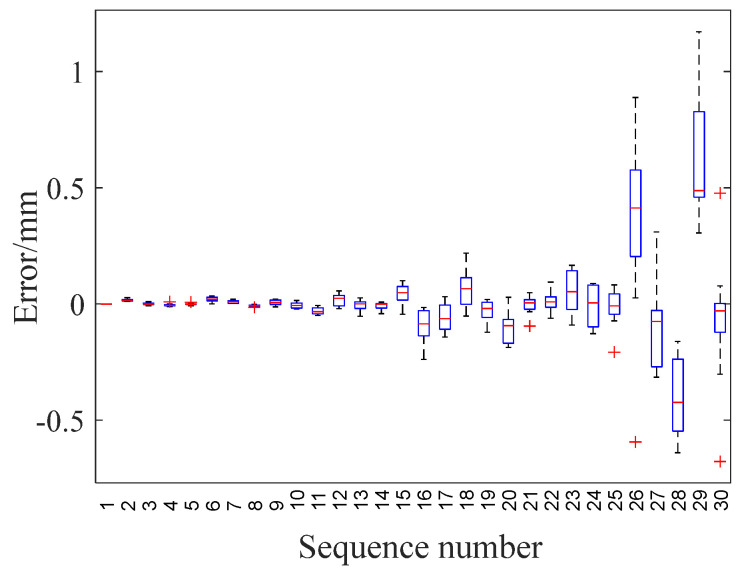
Error analysis of fuzzy logic thickness measurement results.

**Table 1 sensors-24-04066-t001:** Comparison of EMAT thickness measurement equipment.

Manufacturer	Frequency (MHz)	Range (mm)	Thickness Measurement Error (mm)
Oktanta EM1401	3.5–5.5	2–200	±0.1 (thickness 2–200)
ACS A1270	3.0	1–100	±0.1
Orisonic ETGmini-X2	4.0	1.5–300	±0.05 (thickness 1.5–10) ±0.01 + H/200 (thickness 10–500)
Ours	0.5–5.0	0.3–1000	±0.03 (thickness 0.3–10) ±0.01 + H/300 (thickness 10–1000)

**Table 2 sensors-24-04066-t002:** Hardware limitations.

Hardware	Limitation	Frequency (MHz)	Problems
Power amplifier	Frequency	0.05–20	High frequency energy attenuation is serious
Weak signal amplifier	Filter	0.5–5	Bandpass filter
DAC	Refresh rate	50	The actual sine wave frequency is lower than 5 MHz
ADC	Sample rate	50	Restricted Nyquist sampling theorem
EMAT’s coil	Impedance matching frequency	3	Narrowband matches are flat around the center frequency

**Table 3 sensors-24-04066-t003:** Fuzzy logic decision rules.

D	Fuzzy Logic
0.3~1.5	S
1.5~2.5	SM
2.5~4.8	M
4.8~6	ML
≥6	L

## Data Availability

Data are contained within the article.

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
