# Peer review of "Thickness Measurements with EMAT Based on Fuzzy Logic"

_sensors, 2024, doi:10.3390/s24134066_

Round 1

Reviewer 1 Report

Comments and Suggestions for Authors

This manuscript presents an investigation of EMAT using fuzzy logic approach. While in general it is well-written, it is not clear to me the novelty of this work. As authors mention, EMAT is a matured technology. Please clarify how these results are novel and how do they compare with existing commercial transducers, in the introduction. Other than I don't have other comments. 

Reviewer 2 Report

Comments and Suggestions for Authors

The article discusses a novel combined thickness measurement method aimed at overcoming limitations in current non-contact ultrasonic techniques, particularly regarding range and accuracy in industrial applications. This method integrates fuzzy logic principles with minimal hardware requirements to enhance measurement capabilities. It utilizes the short pulse time-of-flight (TOF) technique for initial thickness estimation and incorporates measurements from resonance, short pulse echo, and linear frequency modulation echo to extend the measurement range and improve accuracy. Rigorous experimental validation confirms the method's effectiveness, demonstrating a measurement range of 0.3-1000.0 mm with a median error within ±0.5mm. Compared to traditional methods like short pulse echo, this approach outperforms in accuracy and holds significant industrial potential, offering broader adoption prospects for electromagnetic ultrasonic (EMAT) technology.

The following questions need to be answered before publication.

1. Kindly avoid group citations and the term 'we' in the manuscript.

2. What are the limitations faced by current non-contact ultrasonic methods in metal thickness measurement?

3. How does the novel combined thickness measurement method aim to address these limitations?

4. Why did the authors select the Fuzzy approach, which has ambiguity?

5. Why not other AI tools not considered?

6. How the fuzzy decision rules are arrived at?

7. How does the integration of measurements from resonance, short pulse echo, and linear frequency modulation echo enhance the method's capabilities?

8. How does the proposed method demonstrate its potential for industrial applications?

9. In Table 3, errors need to be reported.

Comments on the Quality of English Language

 Moderate editing of the English language is required

Round 2

Reviewer 2 Report

Comments and Suggestions for Authors

Congrats to the authors.

Comments on the Quality of English Language

 Minor editing of the English language is required.